# The Impact of Polychlorinated Biphenyls on the Development of Zebrafish (*Danio rerio*)

**DOI:** 10.3390/biomedicines12092068

**Published:** 2024-09-10

**Authors:** Megan Moma, Abi Lee, M. Brady Olson, Karin L. Lemkau, W. James Cooper

**Affiliations:** 1Biology Department, Western Washington University, 516 High St., Bellingham, WA 98225, USA; megelise1@gmail.com (M.M.); leea81@wwu.edu (A.L.); olsonm@wwu.edu (M.B.O.); 2Marine and Coastal Science Program, Western Washington University, 516 High St., Bellingham, WA 98225, USA; lemkauk@wwu.edu; 3Chemistry Department, Western Washington University, 516 High St., Bellingham, WA 98225, USA

**Keywords:** zebrafish, development, metamorphosis, feeding efficiency, polychlorinated biphenyls, PCBs, endocrine disruptor, legacy contaminant

## Abstract

Polychlorinated biphenyls (PCBs) are a group of 209 highly stable molecules that were used extensively in industry. Although their commercial use ceased in 1979, they are still present in many aquatic ecosystems due to improper disposal, oceanic currents, atmospheric deposition, and hydrophobic nature. PCBs pose a significant and ongoing threat to the development and sustainability of aquatic organisms. In areas with PCB exposure high mortality rates of organisms inhabiting them are still seen today, posing a significant threat to local species. Zebrafish were exposed to a standard PCB mixture (Aroclor 1254) for the first 5 days post fertilization, as there is a gap in knowledge during this important developmental period for fish (i.e., organization of the body). This PCB mixture was formally available commercially and has a high prevalence in PCB-contaminated sites. We tested for the effects of PCB dosage (control (embryo water only; 0 mg/L), methanol (solvent control; 0 mg/L); PCB 1 (0.125 mg/L), PCB 2 (0.25 mg/L), PCB 3 (0.35 mg/L), and PCB 4 (0.40 mg/L)) on zebrafish survival, rate of metamorphosis, feeding efficiency, and growth. We found significant, dose-dependent effects of PCB exposure on mortality, feeding efficiency, and growth, but no clear effect of PCBs on the rate of zebrafish metamorphosis. We identified a concentration in which there were no observable effects (NOEC), PCB concentration above the NOEC had a significant impact on life-critical processes. This can further inform local management decisions in environments experiencing PCB contamination.

## 1. Introduction

Polychlorinated biphenyls (PCBs) are a group of 209 highly stable molecules that were first developed in 1929 [1]. They were extensively used in machinery such as dielectric fluid capacitors and closed-system heat exchangers as they have a high boiling point, good insulating properties, low flammability, and are chemically stable at high temperatures [1,2]. Within the United States, the Monsanto Company was the only producer of PCBs. They manufactured eight different commercial preparations trademarked as Aroclors [1]. Although sales were restricted to ensure controlled disposal, improper disposal, river and ocean currents, atmospheric deposition, and their hydrophobic nature, have resulted in PCBs being prevalent in many aquatic ecosystems where they pose a significant and ongoing threat [3,4,5]. Due to their massive environmental impact (i.e., the high mortality of exposed organisms), their commercial use was prohibited worldwide in 1979 [1,6].

PCBs are often described as “legacy contaminants” because of their slow rate of degradation and consequently long environmental lifetimes. The prevalence of PCBs in some sediments and their propensity to bioaccumulate makes them a particular threat to species dwelling on the sea floor and those that feed at high trophic levels [7,8]. Furthermore, organisms inhabiting aquatic habitats close to heavily industrialized areas face the highest risk from PCB contamination [9].

Human exposure to PCBs can result in health concerns that range from minor to lethal (e.g., cancer, periorbital edema, gingival hyperplasia, abnormal skull calcification, low birth weight, etc.) [10,11]. The International Agency for Research on Cancer has classified PCBs as a potential carcinogen in humans [10]. Washington Department of Health has advocated against human consumption of Chinook Salmon from Puget Sound due to high levels of PCB contamination [12]. Consuming contaminated seafood is one of the most common paths of human PCB exposure, while inhalation and absorption through the skin can also occur [13,14,15,16,17,18].

In fishes, PCB exposure and subsequent accumulation occur via two main pathways: prey consumption and uptake via gills, epithelial, and dermal tissues [19,20,21,22]. Once exposure occurs, PCBs can significantly alter development and impair mechanisms of homeostasis [23]. Many of these effects are the result of PCBs acting as endocrine-disrupting compounds [24]. PCB exposure can, for example, reduce levels of circulating thyroid hormone (TH) in vertebrates by as much as 30% [3,24,25,26].

Fishes are most sensitive to environmental pollutants during early development [27]. When exposed to PCBs as embryos, fish are more likely to suffer from long-lasting effects due to the impact that PCBs can have on processes that coordinate anatomical organization, such as TH signaling [27]. Adequate TH levels are necessary for fish to metamorphose, build and maintain their skeletons, and develop functional adult feeding mechanisms [28,29]. Decreased TH levels (hypothyroidism) result in the abnormal retention of cartilaginous regions within the skull vault and incomplete skull ossification, leading to decreased levels of cranial motion that can impair feeding [28,29,30,31].

Pollutants stress living systems and can exacerbate the mortality that would normally occur during complicated developmental transitions, such as metamorphosis [32]. Because metamorphosis is associated with high mortality under normal conditions it is sometimes referred to as a “bottleneck” period [32]. Any toxins that further increase metamorphic mortality can have a large impact on population survival [32].

During development, an organism spends a great deal of energy on growth, with a small margin for energy that can be used for other biological processes without impacting survival [33]. Metamorphosis requires significant energy expenditure, so exposure to toxins that interfere with metamorphosis may incur an energetic cost that can lead to mortality [32,33]. Within the field of toxicology, it is common for experiments to focus on PCB exposure during early life stages (especially the embryonic period), while ignoring later developmental periods that may be particularly susceptible to PCB toxicity. The high energetic cost of metamorphosis in combination with the fact that PCBs can disrupt the TH signaling that initiates and directs this important developmental transition suggests that this developmental stage could be heavily impacted by PCB exposure.

Further investigations are needed to better understand how PCBs affect development and fitness, especially during later life stages (e.g., metamorphosis and juvenile development). Although PCB contamination is slowly dwindling, it will continue to pose an environmental threat for many decades. Studies that provide additional insight into the effects of PCBs will allow for the development of better environmental monitoring methods and a clearer understanding of how PCB contamination that is sub-lethal to embryos can have large effects on fish. Juvenile zebrafish (after metamorphosis, but not sexually mature) exposed to PCBs display muscle dysfunction, swimming defects, disruption of liver metabolism, and decreased reproductive fitness [6]. This study provides a better understanding of the effects of PCB contaminants on a model aquatic species, the zebrafish (*Danio rerio*) [34].

We determined threshold PCB tissue concentrations for significant impacts on the development of young zebrafish and quantified the effects of PCB exposure on their growth, survival, rate of metamorphosis, and feeding ability.

Aroclor 1254 (~21% C_12_H_6_C_l4_, ~48% C_12_H_5_C_5_, ~23% C_12_H_4_C_l6_, ~6% C_12_H_3_C_l7_) is the commercial PCB mixture that was used in this study [35]. It is 54% chlorine by mass (as denoted by the last two digits in its name) [35,36,37,38,39]. We chose to examine the effects of Aroclor 1254 on zebrafish development because of its high prevalence in PCB-contaminated sites and due to it having been one of the most widely used PCB mixtures [37]. This research will assist conservation biologists and aquatic resource managers in determining when PCB contamination represents a significant risk to fish stocks.

This study aims to answer four main questions: 1. What PCB concentrations significantly affect survival? 2. What PCB concentrations significantly affect the timing of metamorphosis in fish being exposed? 3. What PCB concentrations significantly affect the feeding efficiency of fish being exposed? 4. What PCB concentrations significantly affect the standard length in fish being exposed? The associated null hypotheses are as follows: PCB exposure will not significantly affect survival, the timing of metamorphosis, feeding efficiency, and standard length.

## 2. Materials and Methods

Wild-type zebrafish (*Danio rerio*; AB line) were used in this study as they are easily bred, have rapid development, and are a model organism with straightforward husbandry [40]. Four male/female zebrafish pairs were placed in each of 4 standard zebrafish breeding tanks (Tecniplast, West Chester, PA, USA) and maintained at 28 degrees C in an incubator (Shel Lab SMI12 incubator, Stellar Scientific, Baltimore, MD, USA) overnight. Incubator lighting was adjusted to the 14:10 light/dark schedule to which the breeding pairs had been previously acclimated. Fish were placed in tanks after lights out to promote fertilization at artificial sunrise the next day.

After breeding, 60 healthy, fertilized eggs were haphazardly selected and added to each of 42 glass petri plates (90 mm diameter; Bomex, Shanghai, China) containing 50 mL of embryo water [41]. This allowed for 7 replicates of 6 treatments (plates as replicates).

Because the PCB mixture was dissolved in methanol, we utilized two control treatments: embryo water alone and embryo water plus methanol (a “solvent control”). We will refer to our treatments in the following manner, with the PCB concentrations of the treatment solutions in parentheses: control (embryo water only; 0 mg/L), methanol (solvent control; 0 mg/L); PCB 1 (0.125 mg/L), PCB 2 (0.25 mg/L), PCB 3 (0.35 mg/L), and PCB 4 (0.40 mg/L). All treatments except for the control treatment and PCB 4 received additional methanol so that the concentration of methanol in all treatments (except for the control) was equal. Treatments are reflective of PCB concentrations found within Pacific herring inhabiting Puget Sound [42]. Eggs remained in these solutions for 5 days and dead eggs were removed daily. By 5 days post-fertilization (dpf) all eggs had hatched. Treatment solutions were removed from each plate via pipette and all larvae were gently rinsed three times with embryo water (pipetted carefully into plates and then pipetted out). PCB exposure ceased at this time. All dead eggs and PCB water waste were disposed of following an approved animal care protocol (WWU 21-006).

After rinsing, the larvae from 6 plates per treatment (36 plates total) were transferred to individual 4 L mason jars (one jar per plate/replicate) containing 500 mL of embryo water. Fish from the remaining petri plates (1 per replicate) were euthanized, for initial length comparisons, according to animal care protocol WWU 21-006, fixed in paraformaldehyde and stored as described below. Jars were assigned to one of six, 110-quart plastic tubs using a random number generator in Excel (Microsoft, Inc., Redmond, WA, USA). Water was placed in the bottom of each tub (~3 inches) to provide a water bath in order to stabilize temperature. A Uniclife 50-watt aquarium heater (Amazon.com, Seattle, WA, USA) was used to maintain the temperature of the water bath at 28 degrees C and an air stone was used to circulate the heated water throughout the tub and maintain an even temperature throughout. The temperature of the water baths was recorded each day and adjusted as necessary.

### 2.1. Daily Care

Eighty percent of the water in each jar (400 mL) was exchanged for new embryo water every day from 6 dpf onward. PCB wastewater was disposed of according to the approved animal care protocol (WWU 21-006). Ammonia levels were measured and recorded daily, 6 dpf onward, for each jar (API NH_3_/NH_4_^+^ Test Kit, API, Chalfont, PA, USA). Ammonia-absorbing sponges (EA Aquatics, San Rafael, Philippines) were cut into 1.5 cm × 1.5 cm squares added to each jar (1 sponge section per jar). Sponges were replaced and changed, starting on 6 dpf, every other day. Jars were inspected daily, and any dead fish were removed. Mortality was recorded daily for every jar. The fish in each jar were fed 50 mL of live *Paramecium* culture once daily after water changes and the removal of any dead fish. Beginning at 10 dpf, 3 drops of live, newly hatched brine shrimp (*Artemia*) were also added to each jar using a transfer pipette.

Brine shrimp were raised in standard brine shrimp cones (Brine Shrimp Direct, Ogden, UT, USA) for 24 h, then allowed to feed on a commercial algal suspension (Reed Mariculture, Campbell, CA, USA) for an additional 24 h. Cultures were passed through a brine shrimp strainer (Brine Shrimp Direct, Ogden, UT, USA), rinsed briefly, and then washed from the strainer into a beaker using DI water. Live brine shrimp were allowed to briefly settle to the bottom of the beaker so that concentrated shrimp could be removed by pipette. The amount of shrimp added to each jar daily was gradually increased at a rate that allowed fish to consume all/most shrimp (following established protocols within the lab). The following day, during water changes, any uneaten shrimp were removed by pipette. At 25 dpf *Paramecia* feeding stopped and fish were only fed 10 drops of brine shrimp once daily for the remainder of the study.

### 2.2. Metamorphosis

Fish were checked daily for signs of metamorphosis starting on 10 dpf (the earliest day at which metamorphosis has been reported in wild-type zebrafish) [43]. Fish were first examined in their jars against a white background in a well-lit area. If any fish appeared to exhibit possible signs of metamorphosis, then the contents of the mason jar were gently decanted into a 2.5 L rectangular tank (Aquaneering Inc., San Diego, CA, USA) for clearer viewing and confirmation of metamorphosis. Fish were considered to have entered metamorphosis when they exhibited a lateral patch of iridophores (a shiny, white patch of skin) immediately behind the head that was flanked dorsally and ventrally by horizontal lines of melanophores (black lines; Figure 1) [43]. Metamorphosed fish were transferred to a separate mason jar within the same tub and the number of fish that had entered metamorphosis was recorded daily for each of the original jars. Each replicate of every treatment had a dedicated jar for fish that had entered metamorphosis.

### 2.3. Feeding Trials

Feeding trials were performed at 15, 25, and 35 dpf to test for an effect of PCB and/or methanol exposure on feeding proficiency. Five (5) fish were haphazardly collected from each mason jar and placed into a single 250 mL beaker containing 200 mL of embryo water at 28 degrees C. Each beaker was placed in a lighted incubator at 28 degrees C for ten minutes to allow fish to acclimate. Twenty-five (25) brine shrimp (5 brine shrimp per fish) were added to each beaker. If there were less than 5 fish alive in a jar, the number of brine shrimp was reduced accordingly to maintain a 5-to-1 shrimp/fish ratio. The water volume in each beaker was also adjusted accordingly. After three (3) minutes ice was added to each beaker to halt feeding and euthanize the fish according to WWU animal care protocol 21-006. The remaining brine shrimp were counted, and the fish from each beaker were placed in labeled tubes in which they were fixed in a paraformaldehyde solution at 4 degrees C for 24 h. After fixation, fish were slowly transferred into 75% ethanol for storage. The standard length of each fish was measured under a stereomicroscope (model MZ10F, Leica Microsystems, Teaneck, NJ, USA) using digital calipers. All remaining fish were euthanized following the final feeding trial at 35 dpf whether or not they were included in a feeding trial.

### 2.4. Statistical Analyses

#### 2.4.1. Survival

An initial Kaplan–Meier analysis was used to test for differences in survival between treatments. Because no significant difference was found between the control and the solvent treatment (*p*-value = 0.49195; Table A1), the PCB treatments were only compared to the solvent control treatment in subsequent analyses. To take population density into account a Cox-Proportional Hazard analysis (CPH) was used to test for differences in survival between the PCB and methanol treatments. CPH (formula below) can account for changes in population density over time, whereas Kaplan–Meier analyses cannot. CPH estimates a survival probability for every treatment and then determines the slope of the survival probability (y) PCB concentration (x) relationship. This slope, representing the relationship of the probability of survival for a given PCB treatment, is termed the Hazard Ratio (HR) for survival [44]. An HR (formula below) was also calculated for population density in order to determine if the number of fish per jar influenced survival. CPH analyses were performed using the ‘coxph’ function to run a fixed-effects Cox model in the ‘survival’ package within R Studio [44,45]. The tubs in which the jars were maintained were treated as a random effect and fixed coefficients were calculated to estimate the effects of treatment and population density on survival.
$$h(X_{i},t)=h_{0}(t)exp\left[\sum_{j=i}^{p}X_{ij}b_{j}\right]$$
$$HR\left(X_{i}\right)=\frac{h\left(X_{i},t\right)}{h_{0}(t)}=exp\left[\sum_{j=i}^{p}X_{ij}b_{j}\right]$$

The ‘cox.zph’ function was used to test the assumption that the Hazard Ratio (HR) was constant throughout the study. A Kaplan–Meier survival plot was used to display daily survival across treatments (Figure 2), with changes in slope indicating when fish deaths occurred [46].

#### 2.4.2. Metamorphosis

CPH was also used to examine the metamorphosis data. Because the metamorphosis HR for these data was not proportional throughout the study, the ‘coxme’ function was used to run a mixed-effects Cox model (fixed effects and random effects). This model is not sensitive to the assumption that the HR is constant over time (taking population density into account) [47]. Population density was accounted for in this model because population density changed each time metamorphosing fish were removed from their original jar. We used the same random effects within the experimental design as noted above (jars nested within tubs; see ‘*Survival*’ section) and fixed coefficients were also treated in the same manner. HR were calculated for both survival and population density.

#### 2.4.3. Feeding Efficiency

Feeding efficiency was measured as the percentage of available shrimp consumed during a trial (Figure 3, Figure 4 and Figure 5). A Negative Binomial Model (NBM), which is a specific version of a Generalized Linear Mixed Model (GLMM), was used to compare feeding efficiency across treatments for all feeding trials (15, 25, and 35 dpf; formula below).
$$P\left(X=k\right)=\frac{\Gamma (k+r)}{\Gamma (k+1)\Gamma (r)}{\left(\frac{\Theta }{\Theta +\mu }\right)}^{r}{\left(\frac{\mu }{\Theta +\mu }\right)}^{k}$$

GLMM merges aspects of both a Generalized Linear Model (GLM) and a Mixed Model, and allows for irregular distributions of data [48]. Both fixed and random effects are accounted for within the GLMM model. Fixed effects differentiate differences between treatments. Random effects within this model are the same as those detailed in the section above titled ‘*Survival*’. An NBM uses a Poisson-Gamma mixture to assess count data and allows for high variance in comparison to the mean [49]. This can accommodate overdispersion when the residual variance is higher than what the model can predict.

NBM analyses were run through the ‘lme4’ package in R Studio using the function ‘glmer.nb’ [50]. The ‘q-q plot’ function in the ‘ggplot’ R package was used to determine if the residuals of the data were normally distributed (an assumption of the NBM) [51,52].

Pairwise comparisons of feeding efficiency between treatments were then performed for each feeding trial using the package ‘emmeans’ [52]. The False Discovery Rate (FDR) correction method [52] was used to adjust *p*-values for multiple comparisons.

#### 2.4.4. Length

A GLMM was used to test for differences in the rate of fish elongation between treatments. The steps of these analyses followed the same order as those described for ‘*Feeding Efficiency*’ above. This model was created using the ‘glmer’ function in the ‘lme4’ package in R Studio [50]. The function ‘emmeans’ was used to run pairwise comparisons between treatments at each time point (5, 15, 25, and 35 dpf) [53]. Data from the PCB 4 treatment were not included in the measurements recorded at 35 dpf as all fish in that treatment had died by that time. Q-Q plot was used to verify that the data met the assumptions of the model, and the FDR correction method was also used to adjust *p*-values (see ‘*Feeding Efficiency*’ above) [53,54].

## 3. Results

### 3.1. Survival

PCB concentration had a significant effect on survival (Table 1). We therefore reject our first null hypothesis; PCB exposure will significantly affect survival. Survival data are visually displayed in Figure 2. At 0 dpf the y-axis is at 1.0 (100% of fish were alive). Neither tub nor jar had a significant effect on survival (Table A2). The HR for survival was significant (*p*-value < 0.0001) 2e and estimated to be 130.4 (Table 1), which indicates that exposure to higher PCB concentrations resulted in higher mortality, with PCB-treated fish 130 times more likely to die relative to control fish. The HR for population density was significant (*p*-value ≤ 0.0001) and estimated to be 1.046 (Table 1), indicating that the mortality of PCB-treated fish and control fish were affected in a similar way by population density.

In general, higher rates of mortality were associated with higher PCB dosages (Figure 1). However, fish in the PCB 2 treatment exhibited survival patterns similar to the control treatment (Figure 1).

### 3.2. Metamorphosis

A marginally insignificant (*p*-value = 0.051) effect of PCB concentration on the rate of metamorphosis was found (Table 2).

We therefore fail to reject our null hypothesis that PCB exposure will significantly affect the timing of metamorphosis. In general, as PCB concentration increased the rate of metamorphosis decreased (Table 2; Figure 2). However, an increased rate of metamorphosis was seen in PCB 2 treatment (Figure 2). The PCB 3 treatment exhibited the slowest rate of metamorphosis (Figure 2).

The PCB 4 treatment was excluded from these analyses as there were only four individuals alive at the onset of metamorphosis and this sample size would not support statistical analysis.

In this analysis, an HR of 1 indicates that the treated group acted the same as the solvent control group (methanol) and an HR > 1 indicates a faster rate of metamorphosis relative to the control. An HR of 1.3045 for population density was found to be significant (*p*-value = 0.000; Table 2), which indicates that, across all treatments, the rate of metamorphosis increased as population density within jars decreased. When population density is not accounted for PCB exposure is seen to significantly slow the rate of metamorphosis (HR = 3.621 × 10^−7^, *p*-value = 4.02 × 10^−5^; Table 2). However, when population density is included in the model the rate of metamorphosis is marginally insignificant (HR = 0.0316, *p*-value = 0.051; Table 2).

### 3.3. Feeding Efficiency

Fish from all PCB treatments except the PCB 2 treatment had significantly lower feeding efficiencies than control fish at 15 dpf (Table 3, pairwise comparisons between treatments at 15 dpf). No PCB treatments exhibited significantly different feeding efficiency relative to control fish at 25 and 35 dpf (Table 3; Figure 3).

**Figure 3 biomedicines-12-02068-f003:**
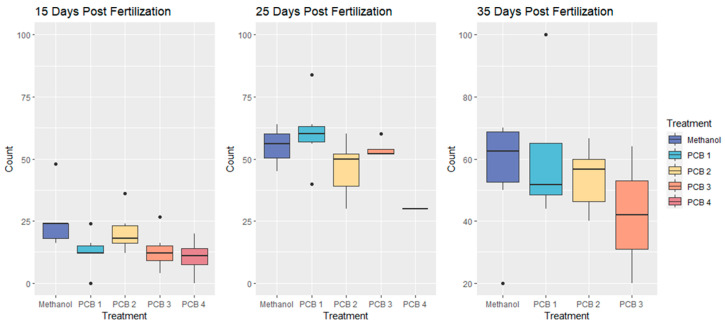
Comparative Feeding Efficiencies of Treatments at 15-, 25-, and 35-Days Post Fertilization (DPF). No PCB 4 fish survived until 35 DPF. Fish from all PCB treatments except the PCB 2 treatment had significantly lower feeding efficiencies than control fish at 15 dpf and no PCB treatments exhibited significantly different feeding at 25 and 35 dpf.

We therefore cannot reject the null hypothesis that PCB exposure will not significantly affect feeding efficiency. However, except for PCB 2 treatment, we found that PCB exposure affected feeding efficiency in younger, pre-metamorphic fish, but that post-metamorphic fish were not strongly affected. The results of the complete pairwise comparisons are included in the supplementary data (Table A3).

Within each treatment, fish at 15 dpf exhibited significantly lower feeding efficiencies than those at 25 and 35 dpf (Figure 4).

**Figure 4 biomedicines-12-02068-f004:**
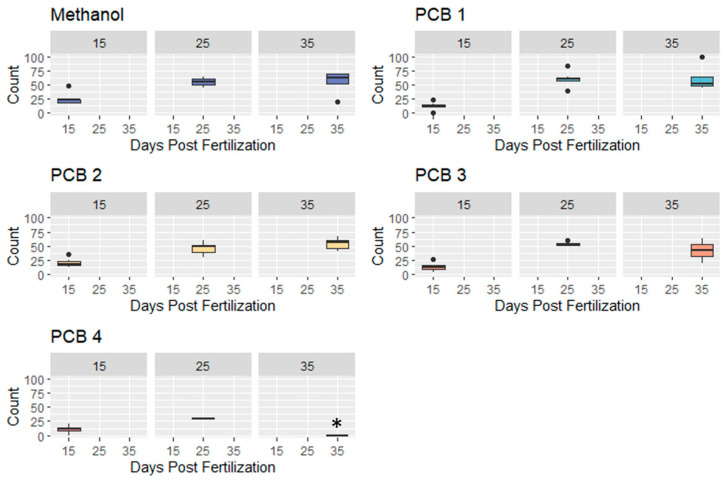
Comparative Feeding Efficiencies within Treatments and Across Development. (* No PCB 4 fish survived until 35 dpf). Fish at 15 dpf had significantly lower feeding efficiencies than those at 25 and 35 dpf (all treatments). There were no significant differences in the feeding efficiencies of 25 and 35 dpf fish within any treatment (Table 4).

There was no significant difference in the feeding efficiencies of 25 and 35 dpf fish within any treatment (Table 4).

Because no fish treated with PCB 4 survived to 35 dpf there could be no comparison with 25 dpf fish from this treatment.

### 3.4. Length

Length distributions were similar across treatments (Table A4) with one exception. At 35 dpf, the surviving fish that had been treated with the highest PCB concentration at the time (PCB 3) had body lengths that were significantly smaller in comparison to 35 dpf fish from the other treatments (Figure 5; Table A4).

**Figure 5 biomedicines-12-02068-f005:**
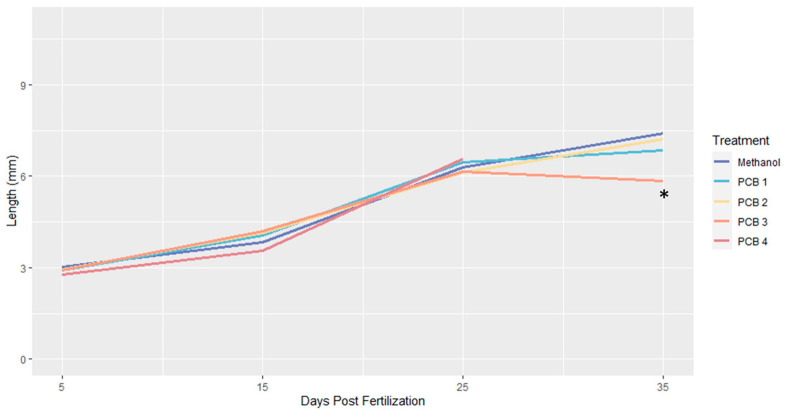
Comparative Average Lengths Across Treatments. No PCB 4 fish survived until 35 Days Post Fertilization (DPF). (* PCB 3 was significantly shorter than other treatments at 35 DPF). See Figure A1 for individual treatment length distributions.

Fish treated with 0.40 mg/L (PCB 4) did not survive to 35 dpf. We therefore cannot reject the null hypothesis that PCB exposure will not affect fish length.

Within treatments, 15 dpf fish were significantly longer than 5 dpf specimens, except for those in the methanol treatment (*p*-value = 0.0561; Table 5), and significantly shorter than both 25 and 35 dpf specimens (Table 5; Figure 5).

In both the methanol and PCB 2 treatments fish underwent significant increases in length between each time point at which length was measured (i.e., 5, 15, 25 and 35 dpf; Table 5). For the PCB 1 and PCB 3 treatments, there were no significant differences in the lengths of 25 and 35 dpf fish (Table 5). Within the PCB 4 treatment, 15 dpf fish were significantly shorter than fish collected at 25 dpf (Table 5).

Among treatments, there were no significant differences in the lengths of fishes collected at the same age except for those collected at 35 dpf (Table A4; Figure 5 and Figure A1). At 35 dpf fish from the PCB 3 treatment were significantly shorter than fish from the other treatments in which specimens survived to 35 dpf (i.e., all other treatments except for PCB 4; Table A4; Figure 5 and Figure A1).

## 4. Discussion

Through this experiment, a threshold value of 0.35 mg/L (PCB 3) was identified at which it is difficult for a fish to successfully undergo metamorphosis. Fish exposed to this threshold showed evidence of decreased survival, prolonged metamorphosis, decreased feeding efficiency in pre-metamorphic stages, and no impact on length until post-metamorphic development. Identifying this threshold is an important step in better informing management plans in exposed ecosystems moving forward locally within the Salish Sea.

Despite nearly half a century of recovery, PCB exposure continues to be a serious threat to aquatic ecosystems [3,4,5]. Our ability to mitigate the effects of this contamination is limited by our understanding of the threshold PCB exposure levels that produce toxic effects in various species, and the manner in which sub-lethal exposure affects life-critical processes in these organisms. The findings reported here contribute to this understanding by identifying threshold levels of PCB exposure that have lethal effects on a model fish (zebrafish) and by examining how sub-lethal exposure impacts their growth, metamorphosis, and feeding ability.

We examined the effects of exposing zebrafish to the most commonly used commercial PCB mixture: Aroclor 1254 [37]. Aroclor 1254 residues are frequently reported in environmental surveys of PCB-contaminated sediments and groundwater [55]. Although we have some understanding of how exposure to Aroclor 1254 affects vertebrates, this study is the first of its kind to quantify the sub-lethal effects of Aroclor 1254 during later developmental periods (e.g., metamorphosis and early juvenile stages). One of the primary mechanisms through which PCBs exert their effect is via TH disruption which then affects metabolism [28,31]. However, the large tissue volumes required to estimate zebrafish TH levels and the cost of the equipment needed to estimate PCB effects on energy expenditure made measuring TH tissue levels and the effects of PCBs/low TH on energy use, etc. unfeasible. Instead, we measured survival and aspects of development both known to be important for survival and to be affected by altering TH levels [27,28,29,30,31].

Because many PCB toxicity studies have focused on determining lethal PCB contamination thresholds during early development [42,56], it is likely that the impact of these compounds has been underestimated. Aquatic organisms that complete their embryonic and larval stages despite PCB exposure may still experience impairment during later life [42]. Reduced growth rates during larval and juvenile development, delayed metamorphosis, and the disorganization of the anatomical remodeling that occurs during metamorphosis, for example, can significantly reduce the survival of aquatic species and heavily impact the annual recruitment of young of the year to existing populations (i.e., stocks) [42,56,57,58,59]. In addition to quantifying the toxic effects of PCBs on early zebrafish development, our findings also improve our understanding of the threats posed by sub-lethal PCB exposure to the sustainability of wild populations.

### 4.1. Impacts of PCB Exposure on Survival

Many organisms have lower abilities to compensate for toxins (e.g., PCBs) during early development relative to later life stages [60]. Exposure to PCBs can also impact the fundamental body organization that occurs during embryogenesis [61]. Because early development is so sensitive to PCB toxicity, we exposed specimens to Aroclor 1254 immediately after fertilization in order to document the most severe effects of this mixture on developing zebrafish.

We found that PCB exposure during early development significantly affected survival throughout embryonic, larval, and early juvenile life stages (Table 1; Figure 2). PCB-treated fish were 130 times more likely to die than methanol-treated fish, with the chance of mortality increasing with PCB concentration (Table 1). These results were consistent with those from similar studies of the effects of PCBs on fish [11,62].

There was higher mortality in early development (0–15 dpf), when fish are generally more susceptible to the effects of PCBs [63] (Figure 2). With the exception of the PCB 2 treatment, we also see a dose-dependent effect of PCB exposure (the strength of the PCB concentration in which fish were immersed) on survival (Figure 2). Most notably, some specimens from every treatment lived until 25 dpf, when zebrafish are normally metamorphosing [64]. The PCB concentrations to which we exposed our specimens were therefore low enough that they did not necessarily prevent zebrafish from reaching the age at which, under normal conditions, they would have completed larval development (i.e., entered metamorphosis).

One of the more important findings of this study is the identification of a threshold PCB exposure concentration for completing fish metamorphosis. In each PCB treatment except PCB 4, we saw specimens that were able to complete metamorphosis and live until 35 dpf (Figure 2). This suggests that exposure to PCB concentrations between 0.35 and 0.40 mg/L (a very narrow range) prevents zebrafish from completing metamorphosis, which is already a developmental transition associated with high mortality in wild fishes [32,65,66,67,68,69,70,71].

Previous studies that identified higher thresholds for the lethal effects of PCBs largely examined embryonic development alone [72,73,74]. Because we also examined later developmental periods, and are able to provide a more accurate estimate of the PCB exposure levels that affect the mortality of developing fishes. This information provides aquatic resource managers with a more accurate threshold for PCB concentrations that will impair the sustainability of wild stocks.

Because PCBs can undergo maternal transfer to eggs and young, as PCBs are transferred to the egg with lipids and proteins [7,73,75,76,77,78,79]. Thus, fishes that develop in PCB-free environments may still be affected by these toxins if their mothers were previously exposed [7]. Maternally inherited pollutants are rarely excreted, and reach peak concentration during the final stages of the yolk-sac embryonic stage of an individual’s life cycle, which can lead to significant developmental disruptions [27].

Threats of PCB contamination to fish stocks are frequently estimated by measuring the PCB concentrations present in the tissues of adult fishes. Our findings suggest that maternal tissue concentrations above 0.35 mg/L (PCB 3) could prevent offspring from surviving to the juvenile stage (Figure 2 and Figure 3) [27,78]. The results of this study therefore provide aquatic resource managers with a useful threshold value that can improve their ability to use current sampling methods (measure of PCB tissue concentrations in adult fishes) to determine if fish stocks are at risk from PCB contamination.

### 4.2. Impacts of PCB Exposure on Metamorphosis

An individual’s survival of metamorphosis is heavily influenced by their energy reserves [80]. Metamorphosis is typically an important “bottleneck period” where post-metamorphic survivors frequently represent a small fraction of the original population [32,65,66,67,68,69,70,71]. Any toxins that impair such a sensitive developmental transition are of strong interest to aquatic resource managers.

PCB exposure has been linked to decreased levels of TH, which plays a significant role in instigating and directing vertebrate metamorphosis [3,81]. Organisms that are unable to maintain satisfactory TH levels may undergo metamorphic delays and/or impairment of the anatomical remodeling that occurs during metamorphosis [28,82,83,84]. Metamorphosis is also physiologically demanding [66,85,86,87,88]. The stress associated with metamorphosis can be exacerbated in polluted environments [89] because many toxins impair this important developmental transition [32,66,83,89]. The presence of Aroclor 1254 in aquatic environments has been shown to prolong the metamorphosis of resident species [6,81,90,91,92,93]. Such elongation of metamorphosis has been shown to increase mortality [94].

Those organisms that survive larval development after PCB exposure may experience prolonged and/or disrupted metamorphosis [62,63,66,81,95]. The metamorphic transition from larva to juvenile is characterized by major transformations in feeding behavior [31]. Significant impacts on the development of post-larval feeding mechanics can therefore result from TH disruption [28,31,82].

PCB exposure delayed metamorphosis in a concentration-dependent manner (HR = 3.621 × 10^−7^, *p*-value = 4.02 × 10^−5^; Table 2; Figure 3). Population density was also found to inversely affect the rate of metamorphosis. When there were fewer fish in a jar metamorphosis occurred more quickly (HR = 1.3045, *p*-value = 0.000; Table 2). Even though lower population densities significantly increased rates of metamorphosis (*p*-value < 0.0001, Table 2), when population density was taken into account, the effect of PCBs on the rate of metamorphosis was still significant (*p*-value = 0.051; Table 2). Methanol and PCB 1 treatment exhibited the most rapid metamorphic rate initially, but this rate leveled off for the remainder of the experiment. PCB 3 showed the slowest metamorphic rate (Table 2; Figure 3). The metamorphosis of fish treated with PCB 2 departed from the pattern established within the other treatments. These fish initially exhibited a rapid rate of metamorphosis and had more individuals entering metamorphosis than in the other treatments (Figure 3).

Although we found that zebrafish exposed to PCB concentrations greater than 0.35 mg/L (PCB 3) did not complete metamorphosis, we did not find a significant effect of PCB concentration on the rate at which the other treatments entered metamorphosis (Table 2; Figure 3). This may be due to the abnormal results seen in PCB 2 (2; Figure 3). This treatment was rerun and the same results were seen. Therefore this concentration may be eliciting a biomechanical response within the fish and will need further research for a definitive conclusion. Visual comparisons of the rates of metamorphosis exhibited by the methanol, PCB 1, and PCB 3 treatments suggest a dose-dependent effect of PCB concentration on the metamorphic rate that would likely place wild populations at risk (Table 2; Figure 3) [96,97,98,99,100].

### 4.3. Impacts of PCB Exposure on Feeding Efficiency

PCBs affected the feeding efficiency of pre-metamorphic zebrafish larvae (15 dpf) in a concentration-dependent manner, but we saw no effects from PCBs on juvenile (i.e., post-metamorphic) fish at 25 and 35 dpf (Table 3 and Table A3). When examining each treatment individually, feeding efficiency was found to be significantly lower at 15 dpf (pre-metamorphic fish) in comparison to 25 and 35 dpf (post-metamorphic fish; Table 3 and Table A3). This supports the conclusion that PCB exposure has a higher impact on larval feeding efficiency than it does on post-metamorphic feeding. Death from larval starvation is common in fishes, and reduced larval feeding can have a strong impact on the seasonal recruitment of young fishes to existing populations [58,96,97,98,99,100]. Even larval fishes that feed sufficiently to survive metamorphosis may experience decreased growth, reproduction, and survival in later life if larval feeding is impaired [42,57].

### 4.4. Impacts of PCB Exposure on Growth

We saw no effects of PCB exposure on growth until after metamorphosis, when the fish in the PCB 3 treatment, the treatment with the highest PCB dosage in which any fish survived past metamorphosis, were found to be significantly shorter than fish from the other treatments (Figure 5; Table 5 and Table A4). There was also a difference in post-metamorphic growth between treatments. Fish in the methanol and PCB 2 treatments exhibited significant growth between days 25 and 35, while PCB 1 and PCB 3 fish showed no significant elongation during this time (Figure 5; Table 5 and Table A4). Fish in the methanol and PCB 4 treatments showed no significant elongation between 5 and 15 dpf (Figure 5; Table 5 and Table A4).

Disruptions in TH signaling are known to retard the growth of young fishes and PCB exposure can reduce TH levels [81]. The possible correlation between slower growth and higher PCB exposure could therefore be the result of lower TH levels in fish treated with higher PCB doses. It should, however, be noted that PCBs affect multiple aspects of development and that these different disruptions could have negative additive effects on elongation [18,74,101,102,103,104]. We cannot therefore definitively attribute the growth reductions seen here to the impact of PCBs on TH signaling alone.

## Figures and Tables

**Figure 1 biomedicines-12-02068-f001:**
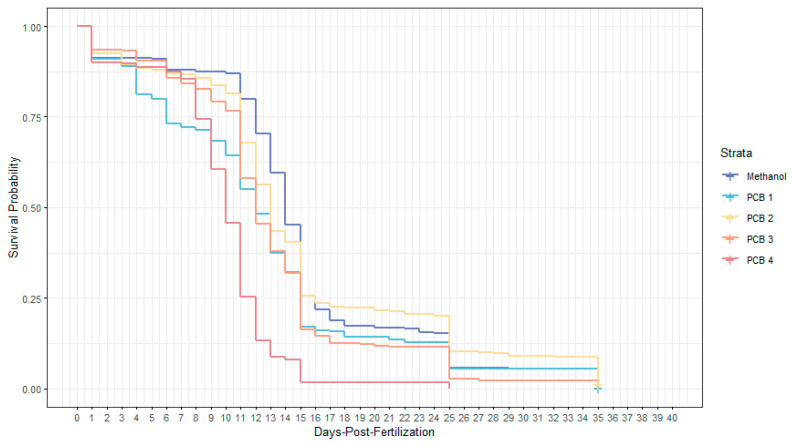
Kaplan Meier Survival Plot Detailing Fish Survival in this Study. Horizontal hash marks at 35 dpf denote individuals that did not die because of PCB exposure but were sacrificed at the end of the experiment.

**Figure 2 biomedicines-12-02068-f002:**
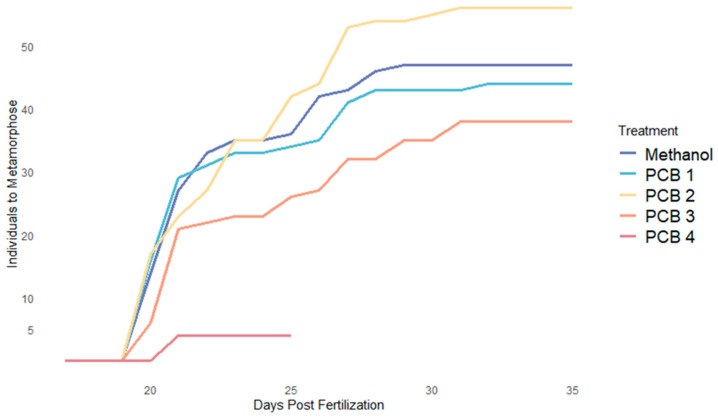
Metamorphosis Across Treatments.

**Table 1 biomedicines-12-02068-t001:** Results of Cox Proportional Hazard Analyses of Survival Data. The estimated effects of PCB concentration and population density on survival are reported.

	COEF	EXP(COEF)	SE(COEF)	Z	PR(>|Z|)
PCB	4.870	1.304 × 10^2^	2.817 × 10^−1^	17.29	<0.0001
Population	4.457 × 10^−2^	1.046	1.181 × 10^−3^	37.73	<0.0001

COEF: Estimated Coefficient, quantifies the effects of each covariate on the Hazard Ratio (HR), EXP(COEF): HR, SE(COEF): Standard Error of COEF, Z: Assesses statistical significance of the COEF, PR(>|Z|): *p*-value.

**Table 2 biomedicines-12-02068-t002:** Results of a Mixed Effects Cox Proportional Hazard Analyses of Metamorphosis Data (with and without population density taken into account). The estimated effects of PCB concentration and population density on the timing of metamorphosis are reported.

	COEF	EXP(COEF)	SE(COEF)	Z	PR(>|Z|)
without Accounting for Population Density					
PCB	−3.454	0.0316	1.7701	−1.95	0.051
Population	0.2658	1.3045	0.0193	13.77	0.000
with Accounting for Population Density					
PCB	1.483 × 10	3.621 × 10^−7^	3.612	−4.106	<0.0001
Population	NA	NA	0	NA	NA

COEF: Estimated Coefficient, quantifies the effect of each covariate on the Hazard Ratio (HR), EXP(COEF): HR, SE(COEF): Standard Error of COEF, Z: Assesses statistical significance of the COEF, PR(>|Z|): *p*-value. Alpha = 0.05.

**Table 3 biomedicines-12-02068-t003:** Results of ‘Emmeans’ Analyses of Feeding Efficiency Data (15, 25, and 35 dpf). FDR-corrected *p*-values are provided for each comparison.

	PCB 1 (15 DPF)	PCB 2 (15 DPF)	PCB 3 (15 DPF)	PCB 4 (15 DPF)	PCB 1 (25 DPF)	PCB 2 (25 DPF)	PCB 3 (25 DPF)	PCB 4 (25 DPF)	PCB 1 (35 DPF)	PCB 2 (35 DPF	PCB 3 (35 DPF)
PCB 2 (15 DPF)	0.0870	-	-	-	-	-	-	-	-	-	-
PCB 3 (15 DPF)	1.000	0.1170	-	-	-	-	-	-	-	-	-
PCB 4 (15 DPF)	0.7255	0.0311	0.1170	-	-	-	-	-	-	-	-
Methanol (15 DPF)	0.0067	0.6247	0.0099	0.0028	-	-	-	-	-	-	-
PCB 2 (25 DPF)	-	-	-	-	0.3429	-	-	-	-	-	-
PCB 3 (25 DPF)	-	-	-	-	0.8026	0.7142	-	-	-	-	-
PCB 4 (25 DPF)	-	-	-	-	0.1516	0.5048	0.2979	-	-	-	-
Methanol (25 DPF)	-	-	-	-	0.8174	0.6247	1.0000	0.2533	-	-	-
PCB 2 (35 DPF)	-	-	-	-	-	-	-	-	0.7376	-	-
PCB 3 (35 DPF)	-	-	-	-	-	-	-	-	0.3770	0.6247	-
Methanol (35 DPF)	-	-	-	-	-	-	-	-	0.8174	1.0000	0.5803

**Table 4 biomedicines-12-02068-t004:** Results of ‘Emmeans’ Analyses of Feeding Efficiency Data Comparing Treatments Across Feeding Trials. FDR-corrected *p*-values are provided for each comparison.

	PCB 1 (15 DPF)	PCB 2 (15 DPF)	PCB 3 (15 DPF)	Methanol (15 DPF)	PCB 1 (25 DPF)	PCB 2 (25 DPF)	PCB 3 (25 DPF)	Methanol (25 DPF)
PCB 1 (25 DPF)	<0.0001	-	-	-	-	-	-	-
PCB 2 (25 DPF)	-	0.0003	-	-	-	-	-	-
PCB 3 (25 DPF)	-	-	<0.0001	-	-	-	-	-
Methanol (25 DPF)	-	-	-	0.0003	-	-	-	-
PCB 1 (35 DPF)	<0.0001	-	-	-	1.0000	-	-	-
PCB 2 (35 DPF)	-	<0.0001	-	-	-	0.6585	-	-
PCB 3 (35 DPF)	-	-	0.0003	-	-	-	0.6432	-
Methanol (35 DPF)	-	-	-	0.0003	-	-	-	1.0000

**Table 5 biomedicines-12-02068-t005:** Results of ‘Emmeans’ Analyses of Length Data for all PCB Treatments. FDR-corrected *p*-values are provided for each comparison.

	PCB 1 (5 DPF)	PCB 2 (5 DPF)	PCB 3 (5 DPF)	PCB 4 (5 DPF)	Methanol (5 DPF)	PCB 1 (15 DPF)	PCB 2 (15 DPF)	PCB 3 (15 DPF)	PCB 4 (15 DPF)	Methanol (15 DPF)	PCB 1 (25 DPF)	PCB 2 (25 DPF)	PCB 3 (25 DPF)	Methanol (25 DPF)
PCB 1 (15 DPF)	0.0072	-	-	-	-	-	-	-	-	-	-	-	-	-
PCB 2 (15 DPF)	-	0.0048	-	-	-	-	-	-	-	-	-	-	-	-
PCB 3 (15 DPF)	-	-	0.0031	-	-	-	-	-	-	-	-	-	-	-
PCB 4 (15 DPF)	-	-	-	0.1124	-	-	-	-	-	-	-	-	-	-
Methanol (15 DPF)	-	-	-	-	0.0561	-	-	-	-	-	-	-	-	-
PCB 1 (25 DPF)	<0.001	-	-	-	-	<0.001	-	-	-	-	-	-	-	-
PCB 2 (25 DPF)	-	<0.001	-	-	-	-	<0.001	-	-	-	-	-	-	-
PCB 3 (25 DPF)	-	-	<0.001	-	-	-	-	<0.001	-	-	-	-	-	-
PCB 4 (25 DPF)	-	-	-	<0.001	-	-	-	-	<0.001	-	-	-	-	-
Methanol (25 DPF)	-	-	-	-	<0.001	-	-	-	-	<0.001	-	-	-	-
PCB 1 (35 DPF)	<0.001	-	-	-	-	<0.001	-	-	-	-	0.0874	-	-	-
PCB 2 (35 DPF)	-	<0.001	-	-	-	-	<0.001	-	-	-	-	<0.001	-	-
PCB 3 (35 DPF)	-	-	<0.001	-	-	-	-	<0.001	-	-	-	-	0.8503	-
Methanol (35 DPF)	-	-	-	-	<0.001	-	-	-	-	<0.001	-	-	-	<0.001

## Data Availability

The data presented in this study are openly available in [Dryad] [DOI: https://datadryad.org/stash/dataset/doi:10.5061/dryad.7sqv9s51j] [Version 1].

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
