# Peer review of "The Impact of Polychlorinated Biphenyls on the Development of Zebrafish (Danio rerio)"

_biomedicines, 2024, doi:10.3390/biomedicines12092068_

Round 1
Reviewer 1 Report
Comments and Suggestions for Authors
Journal: biomedicines
Title: The Impact of Polychlorinated Biphenyls on the Development of Zebrafish (Danio rerio) (3105972)
Comments: Polychlorinated biphenyls (PCBs) are the concerned environmental pollutants for decades. The ubiquity of PCBs in aquatic ecosystem might pose significant threat on the aquatic organisms. Fishes are sensitive to PCBs at early development as PCBs act as endocrine-disrupting compounds. Meanwhile, PCBs were convinced to further increase metamorphic mortality of fish embryos. However, the adverse effects posed by PCBs on the later life stages (e.g., metamorphosis and juvenile development) of fishes are not well-understood. Hence, this study investigated the threshold of PCBs using developmental biomarkers as endpoints. The results indicated that PCBs exposure significantly affected the survival of fish at different development stages, but there was no prevent on the age entering metamorphosis of zebrafish. The threshold of PCB exposure concentration for completing fish metamorphosis is between 0.35 and 0.40 mg/L. The experiment design is valid, and the results are significant. There are some questions should be addressed before publication.
(1) Abstract: the absolute data about the exposure concentrations and the threshold for each biomarker should be added in this section; the environmental implications of this study is insufficient in this section, and please add the associated descriptions for the ecological risks of PCBs assessment.
(2) Introduction: the authors depicted the endocrine effects of PCBs, and the changes of TH levels, energetic cost induced by PCBs exposure. However, the associated indicators were not tested in this study. Actually, the biomarkers linked to the development and the behaviors were as the endpoints in this study. Hence, it’s better to introduce the implications of the developmental and physiological biomarkers for PCBs toxicity assessment.
(3) Materials and Methods: did the exposure go on after the larvae transferred to the fresh mason jars? The descriptions at current form were confusing, and please make it clearer.
(4) Statistical Analyses: the Cox-Proportional Hazard analysis and the Negative Binomial Model were used in this study to analyze the effects posed by PCBs, but the formula was missing. Please add them.
(5) Results: the absolute data combined with the statistical analysis results might be better than only the significant testing results shown, including the survival rates, individuals to metamorphose, feeding efficiencies, and body length; the statistical analysis results should be shown into all figures.
(6) Discussion: Why didn’t the authors test the changes of the contents of TH and the rates energy cost of the fishes exposed to PCBs in this study since the two indicators are so crucial?
Comments on the Quality of English LanguageThere are some minor grammar flaws in the manuscript, and please check and correct.
Author Response
Title: The Impact of Polychlorinated Biphenyls on the Development of Zebrafish (Danio rerio) (3105972)
The authors wish to thank the reviewers for their kind efforts.
Reviewer 1 Comments:
Polychlorinated biphenyls (PCBs) are the concerned environmental pollutants for decades. The ubiquity of PCBs in aquatic ecosystem might pose significant threat on the aquatic organisms. Fishes are sensitive to PCBs at early development as PCBs act as endocrine-disrupting compounds. Meanwhile, PCBs were convinced to further increase metamorphic mortality of fish embryos. However, the adverse effects posed by PCBs on the later life stages (e.g., metamorphosis and juvenile development) of fishes are not well-understood. Hence, this study investigated the threshold of PCBs using developmental biomarkers as endpoints. The results indicated that PCBs exposure significantly affected the survival of fish at different development stages, but there was no prevent on the age entering metamorphosis of zebrafish. The threshold of PCB exposure concentration for completing fish metamorphosis is between 0.35 and 0.40 mg/L. The experiment design is valid, and the results are significant. There are some questions should be addressed before publication.
Comment: Abstract: the absolute data about the exposure concentrations and the threshold for each biomarker should be added in this section; the environmental implications of this study is insufficient in this section, and please add the associated descriptions for the ecological risks of PCBs assessment.
RESPONSE: This change has been made. Please see P 1, L 17,18, 22-24
Comment: Introduction: the authors depicted the endocrine effects of PCBs, and the changes of TH levels, energetic cost induced by PCBs exposure. However, the associated indicators were not tested in this study. Actually, the biomarkers linked to the development and the behaviors were as the endpoints in this study. Hence, it’s better to introduce the implications of the developmental and physiological biomarkers for PCBs toxicity assessment.
RESPONSE: This point is addressed on P 2, L 60-89
Comment: Materials and Methods: did the exposure go on after the larvae transferred to the fresh mason jars? The descriptions at current form were confusing, and please make it clearer.
RESPONSE: It did not go on. Now clearly stated on P 3, L 141
Comment: Statistical Analyses: the Cox-Proportional Hazard analysis and the Negative Binomial Model were used in this study to analyze the effects posed by PCBs, but the formula was missing. Please add them.
RESPONSE: The formulas used are now on pages 5 and 6.
Comment: Results: the absolute data combined with the statistical analysis results might be better than only the significant testing results shown, including the survival rates, individuals to metamorphose, feeding efficiencies, and body length; the statistical analysis results should be shown into all figures.
RESPONSE: We wanted to simplify the tables/figs considerably, as we agree with Reviewer 2’s second comment. We hope that the format is now more legible.
Comment: : Why didn’t the authors test the changes of the contents of TH and the rates energy cost of the fishes exposed to PCBs in this study since the two indicators are so crucial?
RESPONSE: The explanation in now on P 15, L 439-444.
Reviewer 2 Report
Comments and Suggestions for Authors
This is an interesting study on aquatic pollution. Please see the following comments
1. It is not clear to me why so old contaminants have been chosen which have been extensively proven to be toxic to aquatic biota. how this research makes this database significantly better?
2. There are too many graphs and figures in the text. either combine some of them (eg as Table 1a, b, c... and/or transfer some of them to supplementray material
3. the stats performed here are in most cases not the common ones (such as ANOVA or non normality counterpart) but still at least in Figures 3 and 4 some way of distiguishing statistically significant differences between groups should be given
Comments on the Quality of English Languagemino corrections
Author Response
Response to reviewer’s comments
Title: The Impact of Polychlorinated Biphenyls on the Development of Zebrafish (Danio rerio) (3105972)
The authors wish to thank the reviewers for their kind efforts.
Reviewer 2 Comments:
Comment: This is an interesting study on aquatic pollution. Please see the following comments
It is not clear to me why so old contaminants have been chosen which have been extensively proven to be toxic to aquatic biota. how this research makes this database significantly better?
RESPONSE: This is now addressed in more detail on P 2 and 3, L 60-100
Comment: There are too many graphs and figures in the text. either combine some of them (eg as Table 1a, b, c... and/or transfer some of them to supplementary material
RESPONSE: This has been changed significantly (and we agree that this was needed, thank you). The tables and figs have been reduced in number. Many were condensed into the tables now in the appendix.
Comment: The stats performed here are in most cases not the common ones (such as ANOVA or non normality counterpart) but still at least in Figures 3 and 4 some way of distinguishing statistically significant differences between groups should be given.
RESPONSE: We have now noted the groups sig. diff. from each other in the captions of Fig. 3 and 4. In both cases almost all groups were not sig. diff., so this was very straightforward.
Round 2
Reviewer 1 Report
Comments and Suggestions for Authors
The author has addressed all the questions I raised and made the appropriate revisions. I have no further questions and recommend accepting and publishing in present form.